# Towards Geometry Problems Solving Employing GPT-4 Vision with Few-Shot Prompting: An Empirical Study of What Matters

## Abstract

Few-shot prompting methods can significantly improve the ability of Large Language Models (LLMs) in mathematical reasoning, including geometry problem solving (GPS). GPT-4 Vision (GPT-4V), as a leading example of LLMs, also demonstrates significant improvements. This tremendous achievement is mainly attributed to prompting methods like "Chain-of-Thought" and "Program-of-Thought," which leverage the in-context learning ability of the model combined with few-shot prompting to solve new problems. Despite the success of these prompting methods, it remains understood what the GPT-4V model learns from the demonstrations that leads to improved performance. In this paper, we evaluated the answering accuracy of GPT-4V with few-shot prompting on five geometric problem datasets and conducted a series of detailed analyses. Firstly, through ablation experiments with valid and invalid demonstrations, we found that the model's performance improvement is not due to the quality of the demonstration, but rather to the input format, output format, and logic and structure of the demonstration. Secondly, by analyzing the reasoning and computational requirements of GPS, and verifying experimental results, we found that GPS tasks emphasize reasoning ability more than computational power. Finally, our analysis of various prompt methods revealed that existing approaches are not effective at improving model performance concerning problem length and geometric shape. Therefore, specialized prompt methods could be designed to enhance the model's performance in these aspects, or fine-tuning the model by adding geometric problem data with longer lengths or mixed geometric shapes could optimize its performance. Overall, developing an LLM that fully adapts to GPS tasks represents a key research direction. The source code will be made available in a GitHub repository.

## 1 Introduction

It is widely consensus that leveraging the reasoning and in-context learning capabilities of large language models (LLMs) Zhang et al. (2023a); Kandpal et al. (2023); Shi et al. (2023); Ye et al. (2023b), combined with few-shot prompting, can significantly improve their performance in mathematical reasoning Yin et al. (2023); Gao & Das (2024); Firdaus et al. (2023); Liu et al. (2023); Wu et al. (2023b). In mathematical reasoning research, geometry problem solving (GPS) Gao et al. (2023a); Chang et al. (2022); Ning et al. (2023); Peng et al. (2023); Sun et al. (2024a) is crucial as it demands higher levels of specialized knowledge and comprehensive skills Lu et al. (2024), showcasing the potential of large language models more effectively Zhang et al. (2024). Therefore, exploring methods to solve geometry problems using LLMs with few-shot prompting, along with an in-depth analysis of key factors, can provide essential guidance and insights for future research in the GPS field.

Currently, we know very little about what LLMs have learned from prompting methods Chen et al. (2023b); Wu et al. (2023a); Wang et al. (2023a); Gao et al. (2023c); Hu et al. (2024). The most direct way for LLMs to improve their ability to solve geometric problems is by providing a small number of examples, which prompts the model to answer new questions based on these examples. The current prompt methods are mainly divided into two categories: "Chain-of-Thought" Wei et al. (2022);

Zhou et al. (2023); Jiang et al. (2023); Wang et al. (2023b); Mondal et al. (2024) and "Program-of-Thought" Chen et al. (2023a); Gao et al. (2023b); Ye et al. (2023a); Imani et al. (2023). The "Chain-of-Thought" methods utilize the model's extensive pre-training parameters ('memory') Li et al. (2023); Zhong et al. (2024) combined with provided demonstrations to guide the model in reasoning the answer step by step Gao et al. (2023b). In contrast, the "Program-of-Thought" methods leverage the model's code generation ability to use external tools for performing complex computations Chen et al. (2023a). While these methods have shown some success in enhancing the model's problem-solving ability, the key factors driving this improvement remain unknown. To deeply analyze the auxiliary effect of prompting methods on LLMs, we research several key issues: Firstly, are valid demonstrations of "Chain-of-Thought" reasoning or "Program-of-Thought" computation important for guiding GPT-4V in GPS tasks? Secondly, are the "Program-of-Thought" computation methods superior to the "Chain-of-Thought" reasoning methods in GPS tasks? Finally, what other aspects need to be considered when using prompting methods to solve geometric problems, beyond the reasoning process and computation methods?

We have conducted a series of detailed experimental designs and analyses to address the aforementioned key issues Wang et al. (2023a); Gao et al. (2023c). First, to evaluate the impact of including only valid demonstrations on model performance, we conducted ablation experiments using eight different prompting methods. Meanwhile, each method included invalid demonstrations, allowing us to assess the model's ability to learn between valid and invalid demonstrations. From the experimental results, we found that the improvement in the model's performance is not related to the value of the demonstration, but rather to the input format, output format, and the logic and structure of the demonstration. Second, to evaluate whether solving geometric problems is important for improving reasoning or computational abilities, we analyzed the reasoning and computational requirements from the domain knowledge and the complexity of the geometric problems themselves. By comparing the experimental results of two types of prompting methods, we found that solving most geometric problems requires a stronger emphasis on reasoning ability, while only a few geometric problems require complex computational power. Therefore, in GPS tasks, "Chain-of-Thought" reasoning methods are superior to "Program-of-Thought" computation methods. Finally, we analyzed the effect of various prompting methods on the problem length and the geometric shapes involved. The analysis of experimental results showed that these prompting methods did not significantly improve these aspects. This indicates that specialized prompting methods could be designed to improve the model's performance in these aspects, or fine-tuning the model can be achieved by adding problem data with longer lengths or mixed geometric shapes to optimize its performance.

Overall, our research and analysis of LLMs with few-shot prompting to solve geometric problems revealed that the model can learn the prompting framework, including the input format, output format, and answering ideas, from the demonstrations. However, the specific answer process still depends on the problem's content and relies on "memory" to generate an answer. Therefore, proposing a large language model that fully adapts to GPS tasks is a key research direction.

## 2 RELATED WORK

### 2.1 CHAIN-OF-THOUGHT METHODS

The Chain-of-Thought methods involve a series of reasoning prompts that divide a problem into multiple intermediate steps, gradually solving each step to obtain the final answer ultimately. Among these methods, **chain-of-thought prompting (CoT)** Wei et al. (2022), as shown in Figure 1 (a), has achieved remarkable results in solving general reasoning tasks. However, its performance diminishes when tackling problems with higher reasoning difficulty. To address this issue, the **least-to-most prompting (LtM)** Zhou et al. (2023) method is proposed. LtM decomposes a complex problem into a series of easier subproblems and solves them sequentially, using the answers from previous subproblems to assist in solving subsequent ones. To clarify the solution process, the **plan-and-solve prompting (PS)** method Wang et al. (2023b) is proposed. PS solves new problems through a series of simple trigger sentences (such as 'give', 'plan', 'calculation', and 'answer'). The standard CoT method follows a roughly linear reasoning approach but often forgets previous intermediate results during the process. The **residual connection prompting (RP)** method Jiang et al. (2023) mitigates this issue by reintroducing the results of previous steps as prerequisites for subsequent steps, thereby reducing the model's tendency to forget.

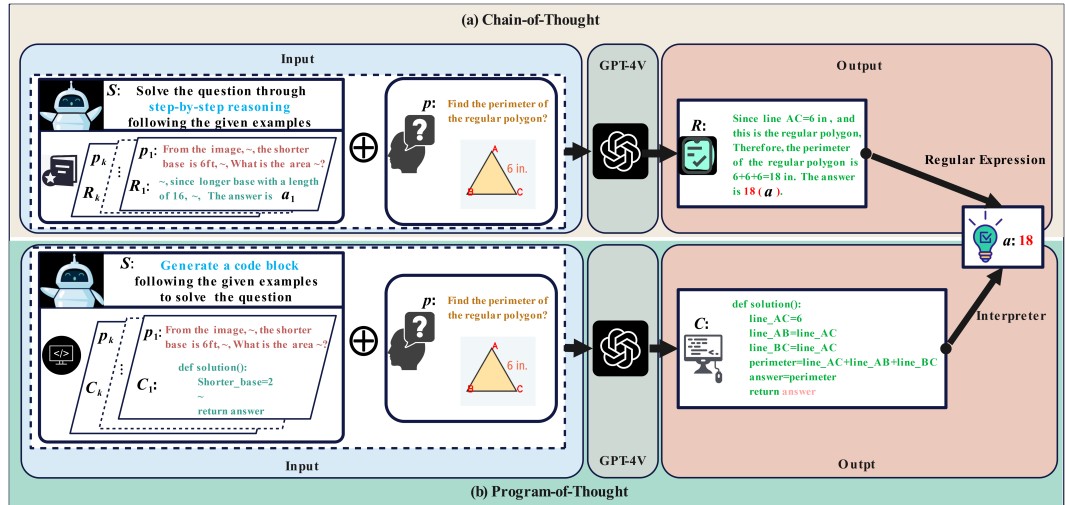

Figure 1: Comparison of the implementation process of two types of prompting methods. (a) Chain-of-Thought methods rely entirely on the GPT-4V' "memory" for step-by-step reasoning and calculation; (b) Program-of-Thought methods generate programs and use external tools for more precise computation.

## 2.2 PROGRAM-OF-THOUGHT METHODS

The Program-of-Thought methods involve a series of computational prompts that utilize the powerful code-generation capabilities of LLMs to solve complex computational problems. Although LLMs excel at gradually decomposing problems using Chain-of-Thought methods, they often encounter logical and arithmetic errors when solving individual subproblems, even if the overall decomposition is correct. To overcome these issues, two representative methods have been proposed: **program-of-thought (PoT)** Chen et al. (2023a) and **program-aided language (PAL)** Gao et al. (2023b), as shown in Figure 1 (b). Both methods separate reasoning from computation by placing each computational step into an external code executor (such as a Python interpreter) for execution. The key difference is that PoT describes each step entirely in the programming language, while PAL integrates both natural language and programming language. To improve the effectiveness in solving search constraint problems, **the satisfiability-aided language (SATLM)** Ye et al. (2023a) is proposed. This method converts natural language reasoning problems into satisfiability (SAT) problems and then obtains answers by using an SAT solver. To enhance the credibility of LLMs-generated answers, **MathPromoter (MP)** Imani et al. (2023) uses hint techniques to generate multiple algebraic expressions and Python functions, solving the same mathematical problem with different approaches. The consensus among these solutions serves as the final answer, thereby increasing the confidence in the output.

## 3 BACKGROUND & STUDY FORMULATION

There is a wide consensus that *Language Models are Few-Shot Learners* Brown et al. (2020); Wu et al. (2023a), and GPT-4V is no exception Sun et al. (2024b); Jin et al. (2024), as reflected in numerous studies. In this section, we first defined two types of prompting methods, Chain-of-Thought Reasoning and Program-of-Thought Computation, and then detailed the input format, output format, and the logic and structure of the demonstration.

### 3.1 CHAIN-OF-THOUGHT REASONING

In the realm of ultra-large-scale unsupervised deep learning, large models are often perceived as black boxes Rai et al. (2023), with their reasoning and decision-making processes being difficult to explain Wang et al. (2023c). This lack of transparency poses a challenge to the credibility of the model results. However, the introduction of the few-shot chain-of-thought method offers a potential

solution Wei et al. (2022); Zhou et al. (2023); Jiang et al. (2023); Wang et al. (2023b); Mondal et al. (2024). This method breaks down logical reasoning problems into multiple steps, generating results with a clear logical context. This approach improves the interpretability of the model, allowing humans to understand the derivation process of the answer. Thus, the chain-of-thought reasoning method marks a crucial milestone in enhancing the success of LLMs.

Specifically, few-shot chain-of-thought reasoning involves LLMs solving a new problem $p$ by following few-shot demonstrations $< p_k \cdot R_k \cdot a_k >$ and generating an answer $a$ with a reasoning process $R$. The standard implementation process is as follows:

$$GPT \{[S, (< p_1 \cdot R_1 \cdot a_1 >, \cdots, < p_k \cdot R_k \cdot a_k >)] \oplus p\} \rightarrow < R, a > \tag{1}$$

where $S$ is a system prompt used to guide the model in generating specific information or completing specific tasks when answering problems. In (a) Chain-of-Thought of Figure 1, we used the system prompt *Solve the question through step-by-step reasoning following the given examples*. Additionally, $k$ represents the number of demonstrations, typically an integer not less than 0. When $k = 0$, it indicates the zero-shot prompting method; when $k = 1$, it indicates the one-shot prompting method; and when $k \geq 2$, it indicates the few-shot prompting method.

$$RE (< R, a >) \rightarrow a \tag{2}$$

Due to the concatenation of reasoning and the answer in the results generated by the LLM, we generally use Regular Expressions ($RE$) to extract the pure answer. For example, in (a) Chain-of-Thought of Figure 1, the output $R$ is: *"Since line AC=6in, and this is a regular polygon, therefore, the perimeter of the regular polygon is 6+6+6=18in. The answer is 18."* Here, the answer $a$ is *18*, and the regular expression we used is: *r "The answer is $(\backslash d+)$ "*.

### 3.2 PROGRAM-OF-THOUGHT COMPUTATION

Even with chain-of-thought reasoning, LLMs do not truly understand mathematical logic or the fundamental concepts of addition, subtraction, multiplication, and division Zhong et al. (2024); Zhou et al. (2024). Instead, they rely on prior knowledge to mimic problem-solving processes, much like "drawing a dipper with a gourd as a model" (a Chinese idiom meaning to imitate without understanding). Therefore, for tasks requiring precise arithmetic, professional computing tools are still necessary, leading to the development of program-of-thought computing Chen et al. (2023a); Gao et al. (2023b); Ye et al. (2023a); Imani et al. (2023).

Specifically, few-shot program-of-thought computation is a method where LLMs solve a new problem $p$ by following few-shot demonstrations $< pk \cdot Ck >$ and generating an answer $a$ using the program $C$ they generate. The standard implementation process is as follows:

$$GPT \{[S, (< p_1 \cdot C_1 >, \cdots < p_k \cdot C_k >)] \oplus p\} \rightarrow C \tag{3}$$

where $S$ is a system prompt designed to guide the model in generating specific information or completing particular tasks when solving problems. In (b) Program-of-Thought of Figure 1, the system prompt *Generate a code block following the given examples to solve the question* is used. Here, $k$ represents the number of demonstrations provided.

$$Interpreter (C) \rightarrow a \tag{4}$$

The core idea of the program-of-thought method for achieving precise arithmetic computation is to input the generated program or code block $C$ into a program interpreter, using professional tools for high-precision computation. For example, in (b) Program-of-Thought of Figure 1, the output $C$ is a *solution* function code block written in Python. We used a Python interpreter to execute this function, obtaining the final answer *18*.

Table 1: Details of datasets being tested. The "total" represents the problem number of questions in an original dataset, and the "sample" represents the number of problems sampled from a dataset in a test.

| Dataset | Total | Sample | Average problem words | Average knowledge |
|---------|-------|--------|----------------------|-------------------|
| GEOS | 186 | 62 | 24.7 | 1.3 |
| Geometry3K | 3002 | 1000 | 12.2 | 1.6 |
| GeoQA | 4998 | 1666 | 52.5 | 2.1 |
| GeoQA+ | 7528 | 2510 | 54.5 | 1.8 |
| PGPS9K | 9022 | 2800 | 17.8 | 1.7 |

### 3.3 RESEARCH QUESTIONS

In the prompting methods discussed above, valid reasoning or program examples are provided as demonstrations to illustrate how GPT-4V derives the generated answer to a new problemWu et al. (2023b); Lu et al. (2024). Despite the impressive performance of various prompting methods in mathematical reasoning tasks, we are interested in exploring the following questions:

- **Q1:** Do valid demonstrations of chain-of-thought reasoning or program-of-thought computation matter for guiding GPT-4V in performing GPS tasks? If not, what does GPT-4V learn to obtain the answer?

- **Q2:** Is the program-of-thoughts computation superior to chain-of-thought reasoning prompting method in GPS task? If not, what characteristics in geometry problems would cause this phenomenon to occur?

- **Q3:** What other aspects need to be considered when using prompting methods to solve geometry problems, besides the reasoning process and computation methods?

## 4 EXPERIMENTAL SETUP

### 4.1 DATASETS & IN-CONTEXT EXEMPLARS

The goal of our experiment is to analyze which factors are important for using prompting methods with few-shot demonstrations to assist LLMs in solving geometric problems. Therefore, the dataset used in our experiment includes five publicly available geometric problem datasets. The detailed introduction is as follows: (1) **GEOS** Seo et al. (2015): the dataset contains simple middle school geometry problems with geometric shapes. (2) **Geometry3K** Lu et al. (2021): the dataset contains numerous geometry problems where semantic information is scarce and most values need to be obtained from images. (3) **GeoQA** Chen et al. (2021): the dataset contains rich semantic information for middle and high school geometry problems. (4)**GeoQA+** Cao & Xiao (2022): the dataset is based on GeoQA, which adds more diverse types of geometry problems and forms an enhanced benchmark dataset. (5) **PGPS9K** Zhang et al. (2023b): the dataset has both fine-grained graph annotations and interpretable solution programs, and a small portion of the dataset comes from Geometry3K. Due to budget considerations, we sample a certain number of questions from the five geometric problem datasets being tested, and the number of samples is shown in Table 1.

Since our testing task only involved solving geometric problems, the five datasets share the same problem-prompting template. The only difference is that the demonstrations of the solving process for the same problem are designed based on different prompt methods, including CoTWei et al. (2022), LtMZhou et al. (2023), PSWang et al. (2023b), RPJiang et al. (2023), PoTChen et al. (2023a), PALGao et al. (2023b), SATLMYe et al. (2023a), and MathprompterImani et al. (2023).

### 4.2 EVALUATION

We employed the GPT-4V API (gpt-4-turbo) [1], the mature GPT-4 Turbo model with vision capabilities, for our experiment. To evaluate the performance of GPT-4V with few-shot prompts, we used two strategies: (1) Average answering accuracy: we randomly sampled problems and tested the

---

[1]https://platform.openai.com/docs/models/gpt-4-turbo-and-gpt-4

model for $M$ rounds, taking the average accuracy as the performance metric. (2) Self-consistency answering accuracy: we evaluated performance with self-consistency decoding Wang et al. (2023d); Weng et al. (2023) by uniformly sampling and aggregating the most consistent answer from the $N$ candidate answers. According to the usual research settings Gao et al. (2023b); Wang et al. (2023d); Weng et al. (2023); Imani et al. (2023), $M$ and $N$ are set to 40, and the temperature $T$ is set to 0.

## 5 How Much Does Valid Demonstration Matter? (Q1)

Intuitively, one of the most critical aspects of a demonstration is its logical validity and sound reasoning. If we provide a demonstration with invalid reasoning steps and computational processes, we would expect GPT-4V to fail to answer properly and potentially experience little to no improvement, or even a decrease in accuracy, compared to standard answering accuracy. This is because we are guiding GPT-4V to answer in the wrong way, which could be detrimental rather than beneficial. To test this intuition, we propose a series of studies where we construct invalid demonstrations for various prompting methods and measure their influence on model behavior.

### 5.1 Constructing invalid demonstrations

We have set valid and invalid prompting demonstrations for eight prompting methods Wang et al. (2023a), as shown in Appendix A. Specifically, valid demonstrations are the standard solving processes that can obtain a standard answer for the demonstration problem; invalid demonstrations are not relevant to the demonstration problem (including numerical and textual descriptions), and the standard answer cannot be obtained.

For example, the demonstration geometry problem in Appendix A is *From the image, you can see that the shorter base of the trapezoid is 6ft, the longer base is 16ft, and the height is 12ft. Find the area of the trapezoid.* For the representative prompting method in the Chain of Thought Reasoning series, CoT prompting method, we have designed a valid solving demonstration for solving the geometry problem based on the method: *In this image, since the trapezoid has a longer base (base1) with a length of 16 and a shorter base (base2) with a length of 6. The height, which is the perpendicular distance between the two bases, is given as 12. To find the area of a trapezoid, you can use the formula: Area = 1/2 * (base1 + base2) * height = 1/2 * (16 + 6) * 12=132. The area of the trapezoid is 132. The answer is 132.* Meanwhile, based on previous research, we have set up a completely invalid demonstration, as shown in Appendix A.

### 5.2 Results & Analysis

**Results.** Figure 2 shows the answering accuracy of GPT-4V with one-shot demonstrations for solving geometric problems under different prompting settings (valid and invalid). From the comparison of the valid-invalid experimental results, it can be seen that invalid prompting settings have both an increase and a decrease compared to valid ones. The most significant increase is LtM with the invalid prompting setting, in the Average answering accuracy of Geometry3K datasets, which increased by 7.2% compared to that with the valid. The most significant decrease is PoT with the invalid prompting setting, in the Average answering accuracy of GeoQA+ datasets, which decreased by 6.3% compared to that with the valid. More detailed experimental data is presented in Appendix C. Additionally, we present GPT-4V's answer accuracy with one-shot demonstrations in Appendix B and with three-shot demonstrations in Appendix D.

**Analysis.** On the one hand, through a comparative analysis of valid and invalid demonstrations, we found that invalid prompting settings have both an increase and a decrease compared to valid ones. This indicates that there is no correlation between the prompting validity and the answering accuracy. In other words, valid demonstrations do not matter for GPT-4V with few-shot prompting in GPS tasks. Moreover, this increase or decrease is not particularly significant, indicating that the influence of valid and invalid demonstrations in guiding GPT-4V are consistent, and proving that GPT-4V has learned the same content from both valid and invalid prompting settings, including input format, output format, and logical thinking. On the other hand, compared to the standard GPT-4V, we found that various prompting methods have obvious improvements in GPS tasks. Moreover, we found that the self-consistency strategy has a higher answering accuracy and a smaller increase or decrease compared to the average evaluation strategy. This situation arises because the self-

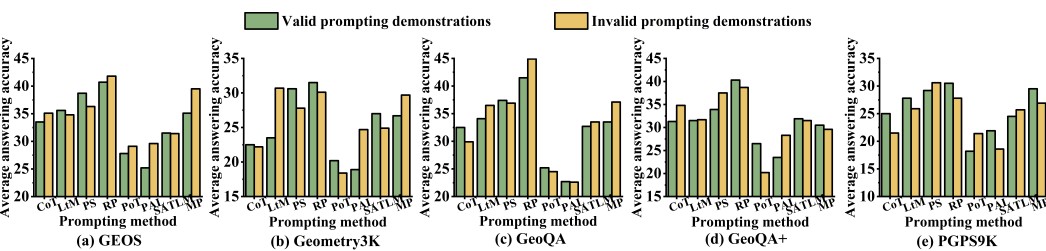

(a) Average answering accuracy of prompting methods with valid and invalid prompting demonstrations

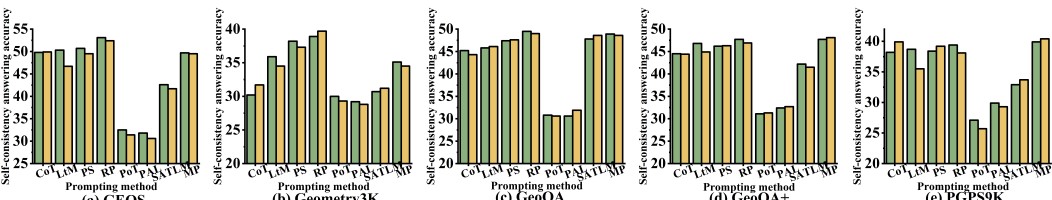

(b) Self-consistency answering accuracy of prompting methods with valid and invalid prompting demonstrations

Figure 2: Comparison of the answering accuracy of GPT-4V employing prompt methods for geometry problems between valid and invalid prompting demonstrations.

consistency strategy is also a prompting method, which guides GPT-4V to find consensus from multiple candidates' answers. This indicates that the self-consistency prompting method is more stable and reliable in the GPS task.

## 5.3 SUMMARY

Based on the experimental results and further analysis in this section, we have summarized two points: in GPS tasks, firstly, the valid demonstrations do not matter for GPT-4V with few-shot prompting, but GPT-4V can learn input format, output format, and logical thinking from the demonstrations. Secondly, although the prompting method significantly improves the performance of GPT-4V, the effects achieved by various prompt methods are different. Among them, the self-consistency strategy overcomes the instability of the model itself to obtain more accurate and reliable answers. However, in GPS tasks, we need to consider more how to improve the reasoning and computing capabilities of GPT-4V. Therefore, we need to further analyze a question: *Is the "Program-of-Thought Computation" superior to the "Chain-of-Thought Reasoning" prompting methods in GPS task? If not, what characteristics in geometric problems would cause this phenomenon to occur?*

## 6 WHICH A SERIES PROMPTING METHOD MATTERS MORE? (Q2)

We tested two types of prompting methods - "Chain-of-Thought Reasoning" and "Program-of-Thought Computation". Intuitively, in GPS tasks, "Chain of Thought Reasoning", which relies solely on memory to solve problems, seems to have no advantage. Instead, the "Program of Thought Computing", which uses tools to enhance performance, can obtain more accurate answers. To test this intuition, we design a series study where we have provided basic evidence for evaluating two types of prompting methods by analyzing the reasoning and computational requirements in GPS tasks.

### 6.1 STATISTICAL REQUIREMENTS

**Reasoning.** The reasoning requirements for solving a geometric problem are positively correlated with the domain knowledge involved in the problem, so, the more domain knowledge the problem involves, the more reasoning steps are required. For example, the domain knowledge involved in problem $p$ in Figure 1 is only one - "Isosceles (Equilateral) Triangle", so the reasoning requirements for the problem are not significant. From the analysis of the average domain knowledge (Average

knowledge) in Table 1, the average domain knowledge of all four datasets exceeded 1.5, indicating the existence of geometric problems involving many domain knowledge, further proving that GPS has a clear reasoning requirement. In Appendix E, we further refined the distribution of domain knowledge involved in the problem. We analyzed and found that problems involving more than 2 domain knowledge accounts for a considerable proportion, with 71.4% in the GeoQA dataset and 40.9% in the GeoQA+ dataset. Specifically, in the GeoQA dataset, 9.4% of the questions involve domain knowledge exceeding 4. This indicates that GPS tasks require complex reasoning to be completed.

**Computation.** The computational requirements for solving geometric problems are related to multi-digit arithmetic. Multi-digit arithmetic refers to arithmetic operations involving numbers. For example, the numerical values involved in solving the problem $p_5$ in Appendix F include 222, and 38707.567, where the largest number 38707.567 is a 5-digit number and the smallest number 222 is a 3-digit number. Therefore, we take 5 (the largest) as the multi-digit. Additionally, we consider the arithmetic computation of decimals numerical values as 0-digit. For example, the numerical values involved in solving the problem $p_0$ in Appendix F include $\frac{4}{7}$, $\frac{5}{7}$, and 0.429, which are regarded as decimals. Therefore, we take 0 (the decimals) as the multi-digit. In Figure 3, we statistically analyzed the distribution of problems with different multi-digit arithmetic in five datasets. We found that the biggest computational requirement in these datasets is also in the five-digit arithmetic, and the vast majority (over 98%) of problems are lower than 3-digit arithmetic. Therefore, the GPS task requires a small amount of computation.

## 6.2 RESULTS & ANALYSIS

**Results.** Figure 2 shows the answering accuracy of GPT-4V with two-shot prompting methods by two evaluation strategies (average and self-consistency answering accuracy ). Firstly, two different background colors represent different prompting methods: the white background in the table represents the "Chain-of-Thought Reasoning" series methods (CoT, LtM, PS, and RP), the gray background represents the "Program-of-Thought Computation" series methods (PoT, PAL, SATLM, and Mathprompter). By comparing the accuracy of these two series methods, it can be seen that the former has a significant advantage in GPS tasks. For example, in the evaluation of the average answering accuracy, the RP method with invalid reasoning (44.9%) on the GeoQA dataset improved the accuracy by 22.3% compared to the PAL method with invalid computation

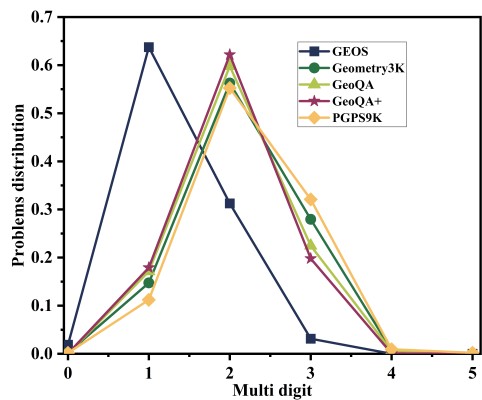

Figure 3: Statistical analysis of computational requirements for different datasets.

(22.6%). Meanwhile, in the evaluation of the self-consistency answering accuracy, the RP method with valid reasoning (53.1%) on the GEOS dataset improved the accuracy by 22.5% compared to the PAL method with invalid computation (30.6%). This indicates that the method of enhancing reasoning ability is more effective for GPS tasks than computation. Furthermore, we also found that the SATLM and Mathprompt prompting methods (belonging to the "Program of Thought Calculation" series methods) exceeded some of the "Chain-of-Thought Reasoning" series methods. For example, in the self-consistency accuracy evaluation of the PGPS9K dataset, the Mathprompt prompting method with invalid computation (40.4%) outperforms all "Chain-of-Thought Reasoning" methods. In addition, by comparing the answering accuracy on two similar datasets (GeoQA and GeoQA+), we found that the accuracy on the GeoQA dataset was lower than that on the GeoQA+ dataset, whether it was the standard GPT (25.4% and 26.5%) or human performance (61.2% and 66.4%). However, after using the prompting method, except for PoT and PAL methods, the accuracy in the GeoQA dataset is generally higher than that in the GeoQA+ dataset.

**Analysis.** On the one hand, according to numerous research analyses, the "Program-of-Thought Computation" series methods are better at handling the problem with large numbers than the "Chain-of-Thought Reasoning" methods, such as *134672 × 98564=?*. But does the GPS task require this

ability? From the analysis in Figure 3, we find that the GPS task requires a small amount of computation. Therefore, there is a phenomenon that the answering accuracy of the "Chain-of-Thought Reasoning" methods is higher than the "Program-of-Thought Computation" series methods. However, it is an exception in SATLM and Mathprompt promotion methods, as these two methods are different from PoT and PAL methods (belonging to the "Program of Thought Calculation" series methods) that rely entirely on the programs for reasoning and computation. On the contrary, they separate the reasoning and computation processes, and the two parts complement each other. Therefore, SATLM and Mathprompt promotion methods surpass some of the "Chain-of-Thought Reasoning" methods. On the other hand, as shown in Appendix E, most problems in the GeoQA dataset require complex reasoning processes, therefore there is a high demand for reasoning ability. Moreover, in the comparison between two similar datasets (GeoQA and GeoQA+) in experimental results, we found that compared to the GeoQA+ dataset, most prompting methods showed a more significant improvement in the GeoQA dataset. This indicates that these methods greatly cater to the complex reasoning requirement of the GeoQA dataset.

### 6.3 Summary

Based on the mutual verification between GPS task requirements and experimental results, we summarize as follows: Firstly, compared to computational requirements, the reasoning requirements are higher. Therefore, the "Chain of Thought Reasoning" series methods (CoT, LtM, PS, and RP) and some "Program of Thought Computing" methods (SATLM and Mathprompter) that guide model reasoning cater to complex reasoning requirements and have more significant improvement effects. Secondly, separating computation from reasoning and using the reasoning process to guide precise computation is an optimal prompting method, such as the SATLM and Mathprompter methods. *Besides reasoning and computational requirements, what other factors can affect the effectiveness of solving geometric problems?*

## 7 WHAT OTHER ASPECTS ALSO MATTER? (Q3)

To analyze whether other aspects besides reasoning and computing requirements would affect the ability of GPT-4V with few-shot prompting to solve geometric problems, we mainly completed two evaluations: geometry problem length and geometry shape.

### 7.1 GEOMETRY PROBLEM LENGTH

The length of geometric problems represents the number of word tokens in the problem text. For example, in Appendix F, $p_0$*: find x.* This problem contains two words, *"find"* and *"x"*, and its length is 2. Intuitively, as the problem length increases, the more semantic information the model needs to understand, the more difficult it is to answer the problem, and the lower the accuracy of the answer. Conversely, the higher the accuracy of the answer. However, in GPS tasks, a lot of information is contained in geometric shapes, and the information contained in the text is limited, resulting in shorter problem lengths, such as $p_3$ in Appendix F. So for solving geometric

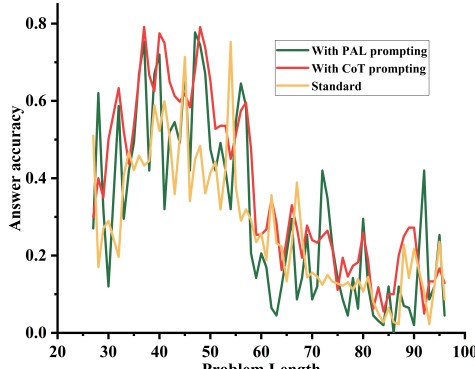

Figure 4: Answering accuracy of GPT-4V with different prompting under different problem lengths, in the GeoQA dataset.

problems, we need to distinguish it from general mathematical reasoning tasks, provide a specific relationship between the problem length and the answering accuracy, and analyze which range of problem length can obtain the optimal answering accuracy.

The answering accuracy of different problem lengths is shown in Figure 4. The experimental results were obtained using the average answering accuracy as an evaluation strategy employing GPT-4V with two-shot PAL and COT at different problem lengths. In addition, to highlight the effectiveness of the prompting method, we also used the standard GPT-4V (without any prompting) as the base-

line. From Figure 4, we can see that regardless of whether there is prompting or not, the accuracy trend in answering at different problem lengths is consistent. And the higher accuracy is concentrated between problem lengths of $(40, 50)$. This indicates that the problem length is unrelated to the method with or without prompting, but only to the model's ability to understand semantic information.

## 7.2 GEOMETRY SHAPE

The fundamental reason why geometric problems differ from general mathematical reasoning problems is that many geometric problems contain abstract geometric shapes, which also pose a huge challenge in GPS tasks - cognitive geometric shapes. To analyze the cognitive ability of GPT-4V towards different geometric shapes, we evaluated the answering accuracy of GPT-4V with different prompting under different geometric shapes in the Geometry3K dataset. The experimental results are shown in Figure 5.

The experimental results were obtained using the average answering accuracy as an evaluation strategy employing GPT-4V with 2-shot PAL and COT at different geometry shapes. In addition, to highlight the effectiveness of the prompting method, we also used the standard GPT-4V (without any prompting) as the baseline. From Figure 5, we can see that GPT-4V has strong cognitive abilities for shapes such as squares, rectangles, and parallelograms. Among them, the accuracy of the CoT prompting method for answering problems involving parallelograms reached 28%.This indicates that the current GPT-4V has good cognitive abilities for simple geometric shapes, but there is still a lot of room for improvement. Furthermore, there is no correlation between the use of prompting methods and the improvement of answering accuracy.

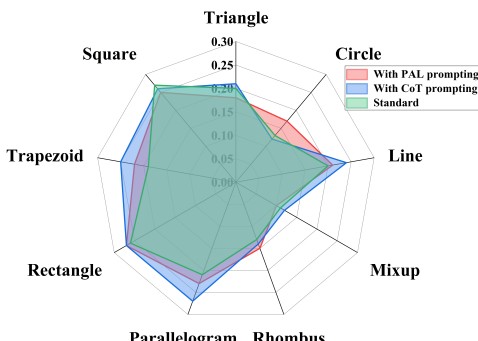

Figure 5: Answering accuracy of GPT-4V with different prompting under different geometry shapes, in the Geometry3K dataset.

## 7.3 SUMMARY

Based on the experimental analysis of geometric problem length and geometric shape, we summarize as follows: Firstly, compared to the standard GPT-4V, the existing prompting methods do not significantly improve the accuracy of answering for a certain problem length or geometric shape. This also provides a starting idea for our future innovative prompting methods. For example, we provide a targeted prompting method for particularly long problems; Alternatively, we can provide visual cues for a certain geometric shape to make it easier for the model to recognize the shape and enhance its cognitive effect. Secondly, the most important thing is to enhance the model's ability to solve geometric problems. Fine-tuning methods are recommended to improve the performance of the model while ensuring that the visual features of geometric shapes and the semantic information of longer problem texts can be fully understood.

## 8 CONCLUSION

In this paper, we aim to better understand what GPT-4V has learned from the few-shot demonstrations, we conducted a series of experiments and detailed analysis. We find that: (1) The model's performance improvement is not due to the quality of the demonstration, but rather to the input format, output format, and the logic and structure of the demonstration; (2) GPS tasks emphasize reasoning ability more than computational power; (3) Specialized prompting methods could be designed to enhance the model's performance. Overall, developing an LLM that fully adapts to GPS tasks represents a key research direction.

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

# A  VALID AND INVALID DEMONSTRATIONS

As shown in Table 2, the rows with a white background represent the Chain-of-Thought Reasoning methods, while the rows with a gray background represent the Program-of-Thought Computation methods. The demonstration problem is: *From the image, you can see that the shorter base of the trapezoid is 6ft, the longer base is 16ft, and the height is 12ft. Find the area of the trapezoid.* And the standard answer is: *132*.

Table 2: A pair of Valid and invalid demonstrations for different prompt methods in our experiment.

| Prompting | Valid | Invalid |
|---|---|---|
| CoT | In this image, since the trapezoid has a longer base (base1) with a length of 16 and a shorter base (base2) with a length of 6. The height, which is the perpendicular distance between the two bases, is given as 12. To find the area of a trapezoid, you can use the formula: Area = 1/2 * (base1 + base2) * height = 1/2 * (16 + 6) * 12=132. The area of the trapezoid is 132. The answer is 132. | The image shows an equilateral triangle, which means all three sides are equal in length. Since one of the sides BC is given as 6 inches, the other two sides (AB and AC) are also 6 inches each. To find the perimeter of the polygon, we sum the lengths of all sides: Perimeter = AB + BC + AC=6+6+6=18. The answer is 18. |
| LtM | Let's break down this problem: 1. What information can we obtain from the graph? 2. How to use this information to calculate the area of this trapezoid? 1. The length of the longer base of this trapezoid is 16, the length of the shorter base is 6, and the height between the longer and shorter base is 12. 2. According to the area formula of the trapezoid, the area of this trapezoid is 0.5 * (16+6) * 12=132. The answer is 132. | A: Let's break down this problem:1.What information can we obtain from the graph? 2. How to use this information to calculate the measure of the perimeter of the regular polygon. 1. A regular triangle with one of the sides, AC, is given as 6 inches 2. Regular triangle, also known as an equilateral triangle, which means all three sides are equal in length. Since one of the sides, AC is given as 6 inches, the other two sides are also 6 inches each. We sum the lengths of all sides: Perimeter = AC + AB + BC=6+6+6=18. The answer is 18. |
| PS | Given: A trapezoidal with a longer base length of 16, a shorter base length of 6, and a height of 12. Plan: We need to calculate the area of this trapezoid. Calculation: the area formula of the trapezoid is 0.5 * (longer base+shorter base) * height; the area of the trapezoid is 0.5 * (16+6) * 12=132. Answer: the area of the trapezoid is 132. The answer is 132. | Given: A regular triangle with one of the sides, AC, is given as 6 inches. Calculation: a regular triangle is also known as an equilateral triangle, which means all three sides are equal in length; since one of the sides AC is given as 6 inches, the other two sides are also 6 inches each. We sum the lengths of all sides: Perimeter = AC + AB + BC=6+6+6=18. Answer: The perimeter of the triangle is 18. The answer is 18. |
| RP | According to the image, the length of the longer base of this trapezoid is 16, the length of the shorter base is 6, and the height between the upper and shorter base is 12. According to the area formula of the trapezoid, the area of the trapezoid is equal to 0.5 * (longer base+shorter base) * height. So the area of this trapezoid is 0.5 * (16[longer base]+6[shorter base]) * 12[height]=132. The answer is 132. | The image shows an equilateral triangle, which means all three sides are equal in length. Since the first side AC is given as 6 inches, the second and third sides (AB and BC) are also 6 inches each. To find the perimeter of the polygon, we sum the lengths of all sides: Perimeter = AC + AB+ BC=6 [the first side]+6 [the second side]+6 [the third side]=18. The answer is 18. |
| PoT | base1 = 16
base2 = 6
height = 12
area = 0.5 * (base1+base2) * height
ans=area | AC=6
AB=AC
BC=AC
Perimeter=AC+AB+BC
ans=Perimeter |
| PAL | # solution in Python:
def solution():
Longer_ base = 16
Shorter_base = 6
Height = 12
Area = 0.5 * (Shorter_base+Longer_ base) * Height
return Area | # solution in Python:
def solution():
Line_AC=6
Line_AB=Line_AC
Line_BC=Line_AC
Perimeter=Line_AC+Line_AB+Line_BC
return Perimeter |
| SATLM | Shorter_base=6
Longer_ base=16
Height=12
Area=Variable()
Area=0.5 * (Shorter_base+Longer_ base) * Height
result=Area
slove(result) | Line_AC=6
Line_AB=Line_AC
Line_BC=Line_AC
Perimeter=Variable()
Perimeter=Line_AC+Line_AB+Line_BC
result=Perimeter
slove(result) |
| MP | Mapping={Shorter_base:6,Longer_base:16,Height:12}
# Algebraic answer
Answer=0.5 * (Shorter_base+Longer_ base) * Height
# python code
def solution(Shorter_base,Longer_ base,Height):
return 0.5 * (Shorter_base+Longer_ base) * Height | Mapping={Line_AC:6,Line_AB:6,Line_BC:6}
# Algebraic answer
Perimeter=Line_AC+Line_AB+Line_BC
# python code
def solution(Line_AC,Line_AB,Line_BC):
Return Line_AC+Line_AB+Line_BC |

# B  ANSWERING ACCURACY COMPARISON OF GPT-4V WITH ONE-SHOT DEMONSTRATIONS

We evaluate the answering accuracy of GPT-4V through the prompting methods with one valid and one invalid demonstration (given in Table 2), respectively. As shown in Table 3, Compared to valid demonstration, prompting methods with invalid demonstration can sometimes improve the accuracy of GPT4-V answers, while at other times they can decrease them. For example, on the GeoQA+ dataset, the PS prompting method with a valid demonstration improved GPT-4V's average answer accuracy by 8.4% compared to an invalid demonstration, whereas it resulted in a 2.2% decrease on the GeoQA dataset. This indicates that the effectiveness of the demonstration is not a factor that affects the performance of the prompting method.

Table 3: Answering accuracy comparison of GPT-4V with one-shot valid and invalid demonstrations for solving geometric problems under different prompting settings on the five benchmark datasets.

| Prompting | Setting | GEOS | Geometry3K | GeoQA | GeoQA+ | PGPS9K |
|---|---|---|---|---|---|---|
| *Average answering accuracy* | | | | | | |
| CoT | Valid Reasoning | 30.2 | 21.5 | 30.5 | 23.1 | 22.1 |
| | Invalid Reasoning | 31.7 | 20.4 | 26.2 | 22.5 | 23.1 |
| LtM | Valid Reasoning | 28.6 | 21.2 | 30.5 | 20.1 | 20.5 |
| | Invalid Reasoning | 30.7 | 23.5 | 26.5 | 25.0 | 20.5 |
| PS | Valid Reasoning | 37.8 | 28 | 32.3 | 30.5 | 23.3 |
| | Invalid Reasoning | 36.2 | 30 | 34.5 | 22.1 | 25 |
| RP | Valid Reasoning | 31.9 | 31.5 | 32.5 | 25.3 | 28.5 |
| | Invalid Reasoning | 37.3 | 30.5 | 38.5 | 28.5 | 22.3 |
| PoT | Valid Computation | 20.1 | 16.2 | 19.3 | 20.5 | 17.6 |
| | Invalid Computation | 21.3 | 15.6 | 20.2 | 20.2 | 11.6 |
| PAL | Valid Computation | 20.8 | 16.4 | 20.3 | 19.3 | 11.9 |
| | Invalid Computation | 19.1 | 17.5 | 21.5 | 18.8 | 11.1 |
| SATLM | Valid Computation | 36.2 | 22.4 | 28.7 | 23.1 | 22.5 |
| | Invalid Computation | 35.3 | 21.9 | 30.9 | 23.3 | 22.7 |
| MP | Valid Computation | 36.2 | 32.5 | 33.5 | 27.4 | 22.3 |
| | Invalid Computation | 42.2 | 34.2 | 36.5 | 29.7 | 23.5 |
| *Self-consistency answering accuracy* | | | | | | |
| CoT | Valid Reasoning | 38.3 | 36.7 | 45.3 | 48.3 | 26.7 |
| | Invalid Reasoning | 37.3 | 36.5 | 46.2 | 47.4 | 27.1 |
| LtM | Valid Reasoning | 41.7 | 40.9 | 43.3 | 34.6 | 33.3 |
| | Invalid Reasoning | 42.3 | 40.1 | 44.7 | 39.7 | 31.6 |
| PS | Valid Reasoning | 43.7 | 31.4 | 37.8 | 35.8 | 25.9 |
| | Invalid Reasoning | 44.5 | 31.9 | 35.3 | 35.3 | 29.1 |
| RP | Valid Reasoning | 45.3 | 26.7 | 40.3 | 40.7 | 33.3 |
| | Invalid Reasoning | 45.5 | 28.1 | 41.4 | 40.4 | 34.7 |
| PoT | Valid Computation | 23.9 | 19.3 | 26.2 | 25.7 | 20.4 |
| | Invalid Computation | 23.6 | 18.7 | 26.9 | 24.1 | 19.8 |
| PAL | Valid Computation | 323.3 | 18.7 | 25.4 | 23.8 | 17.5 |
| | Invalid Computation | 22.9 | 17.4 | 25.9 | 23.5 | 17.8 |
| SATLM | Valid Computation | 33.9 | 31.7 | 39.5 | 38.4 | 31.7 |
| | Invalid Computation | 34.7 | 30.1 | 39.2 | 38.7 | 32.4 |
| MP | Valid Computation | 48.3 | 35.6 | 45.1 | 43.4 | 28.5 |
| | Invalid Computation | 48.3 | 35.7 | 45.6 | 43.9 | 28.3 |

# C ANSWERING ACCURACY COMPARISON OF GPT-4V WITH TWO-SHOT DEMONSTRATIONS

We evaluate the answering accuracy of GPT-4V through the prompting methods with two valid and two invalid demonstrations, respectively. As shown in Table 4, Compared to valid demonstration, prompting methods with invalid demonstration can sometimes improve the accuracy of GPT4-V answers, while at other times they can decrease them. For example, on the Geometry3K dataset, the CoT prompting method with a valid demonstration increased GPT-4V's average answer accuracy by 1.3% compared to an invalid demonstration, whereas it resulted in a 3.5% decrease on the PGPS9K dataset. This indicates that the effectiveness of the demonstration is not a factor that affects the performance of the prompting method.

Table 4: Answering accuracy comparison of GPT-4V with two-shot valid and invalid demonstrations for solving geometric problems under different prompting settings on the five benchmark datasets.

| Prompting | Setting | GEOS | Geometry3K | GeoQA | GeoQA+ | PGPS9K |
|---|---|---|---|---|---|---|
| *Average answering accuracy* | | | | | | |
| CoT | Valid Reasoning | 33.5 | 22.5 | 32.5 | 31.3 | 21.5 |
| | Invalid Reasoning | 35.1 | 21.2 | 29.9 | 34.8 | 25.0 |
| LtM | Valid Reasoning | 35.6 | 23.5 | 34.1 | 31.5 | 27.8 |
| | Invalid Reasoning | 36.8 | 30.7 | 36.5 | 31.7 | 25.9 |
| PS | Valid Reasoning | 38.7 | 30.6 | 37.4 | 33.9 | 29.2 |
| | Invalid Reasoning | 36.3 | 27.8 | 36.9 | 37.5 | 30.6 |
| RP | Valid Reasoning | 40.7 | 31.5 | 41.5 | 40.3 | 30.5 |
| | Invalid Reasoning | 41.8 | 30.1 | 44.9 | 38.7 | 27.8 |
| PoT | Valid Computation | 27.8 | 20.2 | 25.2 | 26.5 | 18.2 |
| | Invalid Computation | 29.1 | 18.4 | 24.5 | 20.2 | 21.4 |
| PAL | Valid Computation | 25.2 | 18.9 | 22.7 | 23.5 | 21.9 |
| | Invalid Computation | 29.6 | 24.7 | 22.6 | 28.3 | 18.6 |
| SATLM | Valid Computation | 31.5 | 27.0 | 32.7 | 31.9 | 24.5 |
| | Invalid Computation | 36.4 | 24.9 | 33.5 | 31.5 | 25.7 |
| MP | Valid Computation | 35.1 | 26.7 | 33.5 | 30.5 | 29.5 |
| | Invalid Computation | 39.5 | 29.7 | 37.1 | 29.6 | 26.9 |
| *Self-consistency answering accuracy* | | | | | | |
| CoT | Valid Reasoning | 49.8 | 30.2 | 45.2 | 44.5 | 38.2 |
| | Invalid Reasoning | 49.9 | 31.7 | 44.3 | 44. 4 | 39.9 |
| LtM | Valid Reasoning | 50.3 | 35.9 | 45.8 | 46.8 | 38.7 |
| | Invalid Reasoning | 46.7 | 34.5 | 46.1 | 44.9 | 35.5 |
| PS | Valid Reasoning | 50.7 | 38.2 | 47.4 | 46.2 | 38.4 |
| | Invalid Reasoning | 49.5 | 37.3 | 47.6 | 46.3 | 39.2 |
| RP | Valid Reasoning | 53.1 | 38.9 | 49.5 | 47.7 | 39.4 |
| | Invalid Reasoning | 52.4 | 39.7 | 49.0 | 46.9 | 38.1 |
| PoT | Valid Computation | 32.5 | 30.0 | 30.8 | 31.1 | 27.1 |
| | Invalid Computation | 31.4 | 29.3 | 30.6 | 31.3 | 25.7 |
| PAL | Valid Computation | 31.8 | 29.2 | 30.6 | 32.4 | 29.9 |
| | Invalid Computation | 30.6 | 28.8 | 31.9 | 32.7 | 29.3 |
| SATLM | Valid Computation | 42.6 | 30.7 | 47.8 | 42.2 | 32.9 |
| | Invalid Computation | 41.7 | 31.2 | 48.6 | 41.5 | 33.7 |
| MP | Valid Computation | 49.7 | 35.1 | 48.9 | 47.7 | 39.9 |
| | Invalid Computation | 49.5 | 34.5 | 48.6 | 48.1 | 40.4 |

# D ANSWERING ACCURACY COMPARISON OF GPT-4V WITH THREE-SHOT DEMONSTRATIONS

We evaluate the answering accuracy of GPT-4V through the prompting methods with three valid and three invalid demonstrations, respectively. As shown in Table 5, Compared to valid demonstration, prompting methods with invalid demonstration can sometimes improve the accuracy of GPT4-V answers, while at other times they can decrease them. For example, on the GeoQA+ dataset, the SATLM prompting method with a valid demonstration increased GPT-4V's self-consistency answer accuracy by 0.3% compared to an invalid demonstration, whereas it resulted in a 0.8% decrease on the GeoQA dataset. This indicates that the effectiveness of the demonstration is not a factor that affects the performance of the prompting method.

Table 5: Answering accuracy comparison of GPT-4V with three-shot valid and invalid demonstrations for solving geometric problems under different prompting settings.

| Prompting | Setting | GEOS | Geometry3K | GeoQA | GeoQA+ | PGPS9K |
|-----------|---------|------|------------|-------|--------|--------|
| *Average answering accuracy* | | | | | | |
| CoT | Valid Reasoning | 36.5 | 32.1 | 35.4 | 33.1 | 23.9 |
| | Invalid Reasoning | 36.2 | 27.5 | 41.1 | 29.2 | 23.5 |
| LtM | Valid Reasoning | 38.4 | 28.5 | 34.5 | 29.5 | 24.5 |
| | Invalid Reasoning | 40.1 | 32.2 | 34.1 | 27.8 | 22.5 |
| PS | Valid Reasoning | 39.2 | 31.7 | 33.8 | 32.5 | 27.5 |
| | Invalid Reasoning | 40.1 | 36.4 | 38.5 | 30.7 | 26.5 |
| RP | Valid Reasoning | 34.1 | 25.9 | 34.6 | 28.8 | 29.9 |
| | Invalid Reasoning | 38.7 | 33.5 | 37.5 | 26.3 | 28.2 |
| PoT | Valid Computation | 28.7 | 22.1 | 26.8 | 26.5 | 20.3 |
| | Invalid Computation | 29.1 | 18.4 | 24.5 | 20.2 | 22.7 |
| PAL | Valid Computation | 25.8 | 19.7 | 26.5 | 25.5 | 25.9 |
| | Invalid Computation | 29.2 | 25.5 | 25.5 | 27.5 | 20.5 |
| SATLM | Valid Computation | 34.6 | 34.7 | 31.5 | 27.5 | 20.6 |
| | Invalid Computation | 36.8 | 27.5 | 34.2 | 29.1 | 27.5 |
| MP | Valid Computation | 32.9 | 34.6 | 35.7 | 28.4 | 24.5 |
| | Invalid Computation | 34.6 | 30.5 | 39.1 | 31.4 | 24.1 |
| *Self-consistency answering accuracy* | | | | | | |
| CoT | Valid Reasoning | 50.4 | 32.3 | 47.5 | 47.1 | 39.6 |
| | Invalid Reasoning | 51.1 | 32.4 | 47.1 | 46.7 | 39.7 |
| LtM | Valid Reasoning | 51.4 | 35.1 | 48.9 | 47.1 | 40.1 |
| | Invalid Reasoning | 52.1 | 35.9 | 48.4 | 46.3 | 40.5 |
| PS | Valid Reasoning | 52.3 | 39.4 | 47.5 | 47.1 | 39.5 |
| | Invalid Reasoning | 52.5 | 40.8 | 46.1 | 47.3 | 40.9 |
| RP | Valid Reasoning | 55.7 | 40.9 | 50.1 | 49.3 | 49.1 |
| | Invalid Reasoning | 55.4 | 40.4 | 50.6 | 49.4 | 39.9 |
| PoT | Valid Computation | 35.7 | 30.4 | 36.8 | 34.3 | 29.5 |
| | Invalid Computation | 35.4 | 30.9 | 36.9 | 33.8 | 29.1 |
| PAL | Valid Computation | 34.8 | 29.6 | 30.7 | 32.9 | 29.3 |
| | Invalid Computation | 34.9 | 30.1 | 30.1 | 32.7 | 30.2 |
| SATLM | Valid Computation | 47.3 | 34.7 | 44.9 | 44.1 | 31.5 |
| | Invalid Computation | 47.2 | 35.9 | 45.7 | 43.8 | 31.9 |
| MP | Valid Computation | 52.9 | 39.5 | 50.4 | 49.3 | 39.9 |
| | Invalid Computation | 53.6 | 39.1 | 50.9 | 49.8 | 38.7 |

# E  THE DISTRIBUTION OF THE DOMAIN-KNOWLEDGE NUMBER

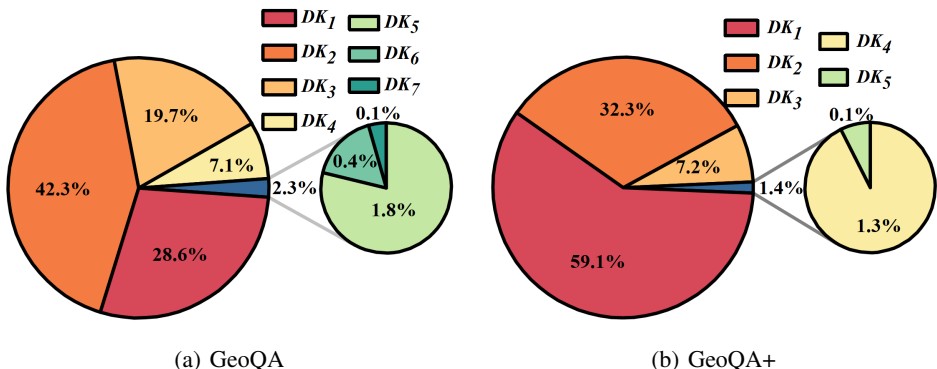

(a) GeoQA

(b) GeoQA+

Figure 6: The distribution of the number of problems involving knowledge from different domains in two datasets, GeoQA and GeoQA+. $DK\_i$ indicates that answering a geometry problem requires at least $i$ domain knowledge.

# F  CASE ANALYSIS OF COMPUTATIONAL REQUIREMENTS

Table 6: Case analysis of computational requirements for different geometry problems.

| ID | Problems | | | Numerical values | Multi-digit |
|---|---|---|---|---|---|
| | text | image | answer | | |
| $p_0$ | Find x. | | 0.429 | $\frac{4}{7}$, $\frac{5}{7}$, 0.429 | 0 |
| $p_1$ | Each pair of polygons is similar. Find the scale factor from polygon ADCB to polygon PSRQ. | | 2.0 | 1.4, 0.7, 2.2, 3.2, 2 | 1 |
| $p_2$ | Find the area of the shaded region. Assume that all polygons that appear to be regular are regular. | | 18.491 | 3, 18.491 | 2 |
| $p_3$ | Find the measure of angle 1. | | 112.0 | 34, 72, 112 | 3 |
| $p_4$ | Find the area of the rhombus. | | 1200.0 | 20, 30, 1200 | 4 |
| $p_5$ | Find the area of the circle. | | 38707.567 | 222, 38707.567 | 5 |

