# OpenReview forum: "Towards Geometry Problems Solving Employing GPT-4 Vision with Few-Shot Prompting: An Empirical Study of What Matters"
_ICLR.cc/2025/Conference — ICLR 2025 Conference Withdrawn Submission_

### Official Review · Reviewer_HL6A · 2024-10-21

**Soundness:** 3
**Presentation:** 2
**Contribution:** 2
**Rating:** 3
**Confidence:** 4

**Summary:**

This paper studies the geometry problem. It observes that the model's performance gain is not due to the quality of the demonstration but to the input format, output format, and logic of the demonstration. Moreover, this analysis finds that specialized prompt methods and find-tuning of the model can optimize its performance.

**Strengths:**

1. This paper targets a hard and interesting mathematical problem called geometry problems. And it compares two series of SOTA promising methods: chain of thoughts and program of thought. It claims that LLMs often "draw a dipper with a gourd as a model" (Ln 194).

2. A wide range of problems are studied, and various analysis experiments are performed.

3. The related works are carefully reviewed, and this analysis paper is well-motivated.

**Weaknesses:**

1. This paper does not propose a specific method to overcome the claimed issues of LLMs, and it does not provide much insight into resolving such problems. In fact, the CoT and PoT methods are well known to the community and are already used in daily life. So it's not clear where the novelty is.

2. Invalid demonstrations would definitely deteriorate the performance of GPT, and this analysis is quite unuseful. What's more concerning is whether the few-shot demonstrations are useful. Note that OpenAI-o1 discourages the use of few-shot demonstrations (refer to their official website).

3. Some claims are quite vague, and the findings cannot support the conclusion. For example, Ln433 concludes that the GPS task requires a small amount of computation. Ln420 suggests that "the method of enhancing reasoning ability is more effective than computation." However, it's not sure whether the proposed computation is the optimal choice of the LLMs. The probability it's the **choice or implementation of CoT / PoT** that hinders the model performance while increasing computation (e.g., making the LLM larger) should significantly improve the model performance.

4. Figure 4 is quite noisy and no meaningful conclusions can be drawn from this figure. Also, it's expected that increasing problem length would deteriorate the accuracy. This analysis does not lead to significant discoveries.

**Questions:**

1. It seems that samples are of the same magnitude as the original dataset, why don't test on the full dataset? It is not clear how the data is sampled from the original dataset.

2. Why not apply the OpenAI-4o model? It's not sure whether the results of this study can be true for the newest OpenAI model.

3. How do you test accuracy given answers? Are they multi-choice questions?

4. In Ln 514, why does the use of prompting methods have nothing to do with the improvement of answering accuracy?

---

### Official Review · Reviewer_zVK4 · 2024-11-02

**Soundness:** 2
**Presentation:** 3
**Contribution:** 2
**Rating:** 5
**Confidence:** 4

**Summary:**

This paper investigates the impact of few-shot prompting methods on enhancing the performance of GPT-4V in solving geometry problem-solving tasks, proposing three key research questions.

The authors first investigate whether valid demonstrations are essential for performance, concluding that prompt structure and logic are more influential than correctness. They then examine whether reasoning (CoT) or computation (PoT) methods are superior for GPS tasks, finding that reasoning-based prompts generally yield better results. Finally, they analyze the influence of various prompting methods on the problem length and the geometric shapes, which all demonstrate minor improvements.

This study suggests that tailored prompting could further optimize GPS performance, paving the way for future research directions.

**Strengths:**

1. Enhancing GPT's ability to solve GPS problems through few-shot prompting is a highly significant topic.
2. The paper is clear and well-structured. It provides a thorough discussion of three key research questions.
3. The paper makes intriguing discoveries: (1) The model’s performance improvement is not due to the quality of the demonstration, but rather to the input format, output format, and the logic and structure of the demonstration; (2) GPS tasks emphasize reasoning ability more than computational power; (3) Specialized prompting methods could be designed to enhance the model’s performance. These findings have the potential to inspire new research directions.

**Weaknesses:**

1. The motivation for studying GPS tasks is not clearly articulated; the authors do not clarify what makes these types of problems uniquely challenging or valuable for research.
2. If I understand correctly, the first research question in the paper has already been thoroughly discussed in previous work. See my questions below.
3. The analysis and discussion of some experimental results are not sufficiently clear or rigorous. See my questions below.

**Questions:**

1. Regarding the first research question, is the difference between the findings of this paper and Wang et al. [1] merely a matter of testing problems?
2. Why does the paper only compare chain-of-thought and program-of-thought? How about tree-of-thought [2] or graph-of-thought [3]?
3. For the reasoning part of the second question, how did you calculate the domain knowledge accounts for each problem? Was it done manually or automatically?
4. Why do you classify problems involving more than two domain knowledge accounts as complex reasoning, while stating that the vast majority of problems, which involve less than three-digit arithmetic, require only a small amount of computation? How do you objectively define what is a high demand for reasoning or computation?
5. What is invalid computation? Could it be that the better performance compared to "invalid computation" (as mentioned in Q1) is due to invalid reasoning providing a standard input-output format, rather than an intrinsic difference between reasoning and computation?
6. For the third research question, what is the significance of analyzing which range of problem lengths yields the optimal answering accuracy?
7. Could the authors clarify the meaning of "the problem length is unrelated to the method with or without prompting, but only to the model’s ability to understand semantic information"?


[1] Wang B, Min S, Deng X, et al. Towards understanding chain-of-thought prompting: An empirical study of what matters. \
[2] Yao S, Yu D, Zhao J, et al. Tree of thoughts: Deliberate problem solving with large language models. \
[3] Besta M, Blach N, Kubicek A, et al. Graph of thoughts: Solving elaborate problems with large language models.

**Details Of Ethics Concerns:**

None.

---

### Official Review · Reviewer_mnyL · 2024-11-04

**Soundness:** 1
**Presentation:** 2
**Contribution:** 2
**Rating:** 3
**Confidence:** 4

**Summary:**

This paper studies the factors that impact the ability of GPT-4 Vision on geometry problem solving (GPS) ability. Experiments are conducted to examine GPT-4’s behavior under various controlled settings. Based on the result, the paper draws the conclusion that (1) the correctness of the demonstrations does not impact the model’s performance; (2) Chain-of-thought outperforms program-of-thought methods, as GPS does not require much computational power from the code-writing; (3) GPT-4V is better at solving problems of shorter description and that concerning simpler shapes, both of which indicate the problem complexity.

**Strengths:**

The study reveals some behaviors of LLMs that can potentially motivate future research.

**Weaknesses:**

1. The study of the GPT-4V under valid or invalid few-shot demonstrations in Section 5 is not straightforward to back up the claim “model’s improvement from few-shot demonstrations is due to input format, output format, and logic and structure of the demonstration” appearing throughout the paper. The study only shows that the overall performance does not degrade by using the demo of the specific invalidity mode in the paper, which is a valid solving process for a wrong problem as shown in Appendix A, compared to using valid demonstrations. It is unclear if it is because “the input format, output format, and logic and structure” is learnt. If there are no demonstrations, is the input format, output format, or logic in the output wrong? If the model is provided with demonstrations of wrong logics (e.g. perimeter = AB * BC *AC), will the model still achieve good performance? The paper can clarify those questions by showing zero-shot GPT-4 (no demonstrations) performance in Figure 2 as comparison, qualitatively comparing the model behavior with no/correct/wrong demonstrations, and testing on more invalidity mode of the demonstrations.

2. While the study in Section 6 shows an interesting result that Program-of-thought (PoT) is outperformed by the Chain-of-thought (CoT), meaning that the code-writing is not suitable for the GPS problem, the analysis of the reason is not convincing. Specifically, the claim is that there are two reasons for this phenomenon: (1) PoT is better at CoT in solving complex arithmetic calculation, but the GPS task does not require much computation; (2) Reasoning in language is better at reasoning in the code. In this case, what is the performance gap between  the two categories of the methods under different calculation complexity and reasoning complexity measurement on the problem level instead of the dataset level? This result will be a stronger support of the claims.

**Questions:**

1. Is Figure 2 the result with one-shot or two-shot demonstration? Section 5.2 claims it is with one-shot, but Section 6.2 claims it is with two-shot. Besides, it is written in Section 6.2 that “...two different background colors represent different prompting methods: the white…, the gray…”. But in Figure 2, there is no distinct background color.

2. What are the “invalid reasoning” and “invalid computation” demonstrations phrased in Section 6.2? Based on the Appendix A, the invalid demonstrations for either method categories are the same, but in language and code format, respectively. When in the code format, it seems they are still valid in computation (no calculation error) but invalid in reasoning (good demo for a wrong problem)?

3. There are also some writing issues in the paper that might mislead the readers:
 - Figure 1, Program-of-Thought method, there is a mismatch between p1 and C1 on the left. “The shorter base is 6 ft” in p1, but it is set to 2 in C1.
- Section 3.2, paragraph 2. “...few-shot demonstration <pk, Ck>...”. ‘k’ should be the subscript.
- Section 6.1, paragraph 1. “In appendix E, we further refined the distribution of…”. The choice of word “refined” is misleading, as it seems Appendix E merely collects the percentage of problems over different knowledge numbers?
- Section 6.2, paragraph 1. “For example, the RP method … improved the accuracy by 22.3% compared to the PAL method …”, “improved” should be “outperforms” as these are two irrelevant methods.

---

### Official Review · Reviewer_Hhu7 · 2024-11-06

**Soundness:** 2
**Presentation:** 2
**Contribution:** 2
**Rating:** 3
**Confidence:** 4

**Summary:**

The paper examines the use of GPT-4V for solving geometry problems through few-shot prompting, assessing how input/output formats, prompt structures, and different reasoning strategies impact performance. It explores two prompting types, Chain-of-Thought and Program-of-Thought, and analyzes their effectiveness across various datasets. Findings suggest that the model’s performance is influenced more by prompt structure than by the validity of demonstrations. Furthermore, reasoning abilities are highlighted as more essential than computation power for geometry problem-solving.

**Strengths:**

1. The reasoning task for LLM is an intriguing and promising area. The authors approach this problem from the perspective of GPS, which presents a fresh and valuable perspective.
2. The experimental evaluation is comprehensive: multiple datasets and prompt types (e.g., CoT, PoT) are used.

**Weaknesses:**

1.	I am confused about the motivation for why we need to answer the three questions the paper is asking (line 55-67). Do these questions really contribute to our understanding of LLM? I feel there is a logical gap between the purpose of understanding LLM and these specific questions. For instance, the question “are valid demonstrations important?” is more about performance tweaks, rather than about actually providing insights on how LLMs work for GPS problems. I cannot directly connect having answers to these performance-related questions to the underlying working schemes of LLMs.
2.	The findings in the paper seem trivial to me. The answers, such as the necessity of valid demonstrations in input/output format, are not surprising. To me, what would actually be interesting is seeing cases where the LLMs can handle bad demonstrations.
3.	There is almost zero algorithmic/technical contribution to this paper. It’s just a bunch of prompts, which any solid paper would have as an ablation study.
4.	The writing quality needs to be significantly improved. For instance, line 74-79 are very vague and poorly explained. There lack of scientific rigor for line 193-195. The claims in line 353-354 are confusing. The conclusion in lines 353–354 regarding the importance of input-output formats,  lacks clear support from preceding paragraphs.
5.	Experimental analysis are weak. Fig. 2 does not demonstrate significant differences across settings, making it difficult to extract meaningful conclusions from the results. The way how the datasets are sampled from the original one is not explained. Some claims are misleading: in line 378, an average domain knowledge score exceeding 1.5 reflects the involvement of extensive domain knowledge actually suggests the opposite of the written claim. Additionally, the notion that the number of digits relates to computational requirements is confusing and inaccurate: it should indicate the precision rather than the computational requirement.

**Questions:**

1.	Please use $``$ rather than $’’$ across the paper
2.	Please use \citep rather than \cite

---

### Note · Authors · 2024-12-13

I have read and agree with the venue's withdrawal policy on behalf of myself and my co-authors.